# Adaptive Safety Early Warning Device for Non-contact Measurement of HVDC Electric Field

**Chunguang Suo [1],\*, Hao Sun [1], Wenbin Zhang [2], Nianrong Zhou [3] and Weiren Chen [2]**

[1]  College of Science, Kunming University of Science, Kunming 650500, China; shwzx@stu.kust.edu.cn
[2]  College of Mechanical and Electrical Engineering, Kunming University of Science, Kunming 650504, China; zwbscg@kust.edu.cn (W.Z.); cwrfjl@stu.kust.edu.cn (W.C.)
[3]  Electric Power Research Institute of Yunnan Power Grid Co., Ltd., Kunming 650011, China; ghlkunm@163.com
*  Correspondence: suochunguang@kust.edu.cn; Tel.: +86-1898-8716-5011

**Abstract:** In order to prevent electric shock accidents during the maintenance and repair of high-voltage direct current (DC) live equipment, an intelligent safety early warning system device based on non-contact electric field measurement is developed in this paper, to ensure the life safety of electrical workers and the stable operation of equipment. Firstly, the self-developed electric field sensor is used to measure the electric field strength of the space around the charged device. The STM32 microprocessor in the sensor will compare the measured electric field value with a preset safety threshold. If the safety threshold is exceeded, the early warning device issues an audible alarm. It prompts the electric power operator to exit to a safe distance. In addition, the alarm prompt symbol and the real-time waveform of the measured electric field can be seen intuitively on the mobile smart APP interface. Secondly, for devices with different voltage levels, the sensor electric field measurement range and sensitivity can be dynamically adjusted by changing the motor speed. In addition, according to the voltage level identification algorithm, this is also to achieve an adaptive safety early warning of different live equipment. Finally, field experiments were performed at a ±500 kV DC converter station and an ±800 kV DC transmission line experimental platform. The measured values of the electric field are basically consistent with the simulated values, and the safety early warning device has no false positives or false negatives during the test. The experimental results show that the design of the safety warning device in this paper meets the actual needs, and it can alert the electric workers to the specified safety distance.

**Keywords:** non-contact measurement; electric field measurement sensor; adaptive adjustment; voltage level identification; security warning

## 1. Introduction

With the widespread application of HVDC transmission technology, it is necessary to regularly maintain and repair the live equipment in HVDC transmission systems to ensure their safe production and application [1]. In order to prevent electric operators from getting an electric shock during the maintenance and repair of DC live equipment, a non-contact DC electric field measurement and safety distance alarm system device is developed for electric operators. This is of great significance to guarantee the life safety of power workers and the stable operation of DC transmission equipment [2,3].

At present, non-contact intelligent early warning devices are mainly used in the field of AC transmission 50 Hz–60 Hz, and the early warning products for high-voltage DC operation scenarios are still in the research stage. In the literature [4,5], there is mentioned a group-based and high-voltage electric field measurement safety distance warning instrument based on wireless transmission.

The real-time waveform of power frequency electric field of transmission line can be monitored by a slave. When the safety distance and the electric field value are exceeded, the slave sends an alarm signal to the host. It realized the dual monitoring of the scene and the command office to prevent power safety accidents. The author proposed and designed a DC electric inspection system based on dual-axis sensing [6], combining with the dual-axis sensing mechanism to improve the vibration capacitance sensor and expand the measurement range of the system. The DDC-01 ground electric field detector and the high-voltage DC combined field strength detection system HDEM-1 and the EFM-115 electrostatic field monitor was developed in the literature [7–9]. They just realized the measurement of the electric field size, about which they do not have a safety warning function, and the measurement system is cumbersome and expensive.

According to the research status of existing technologies, this paper proposes an adaptive security early warning technology solution based on non-contact electric field measurement. The writing idea of the thesis is roughly divided into three steps. The first section introduces the non-contact measurement method of high-voltage DC electric field and the measurement principle of the sensor based on Gauss's theorem. The second section details the design of the induction device in the mechanical structure of the self-developed electric field sensor. The sensor signal processing circuit includes differential amplification, peak detection, low-pass filter, motor drive and electric field polarity discrimination. Additionally, the sensor performance parameters were determined through calibration experiments. The third section introduces how to implement specific technical schemes and voltage level identification algorithms for safety early warning. Finally, field experiment tests were performed at a ±500 kV DC converter station and a ±800 kV DC transmission line experimental platform in Yunnan.

## 2. Measurement Method and Principle

This paper uses a non-contact method to measure the electric field strength in the space around high-voltage DC charged equipment. The principle of the sensor's measurement of the DC electric field is based on Gauss's theorem.

### 2.1. Electric Field Measurement Method

The electric field measurement sensor is placed within a certain distance from the DC-charged device, and the sensor is used to measure the surface potential of the charged device. In essence, the surface electric field of the DC-charged device is measured. The non-contact measurement results are less affected by the meter's input capacitance and input resistance, which is 15% more accurate than contact measurements. But the measurement distance and the geometry of the charged device will affect the measurement results [10]. The non-contact and long-distance measurement method of high-voltage DC electric field strength is shown in Figure 1.

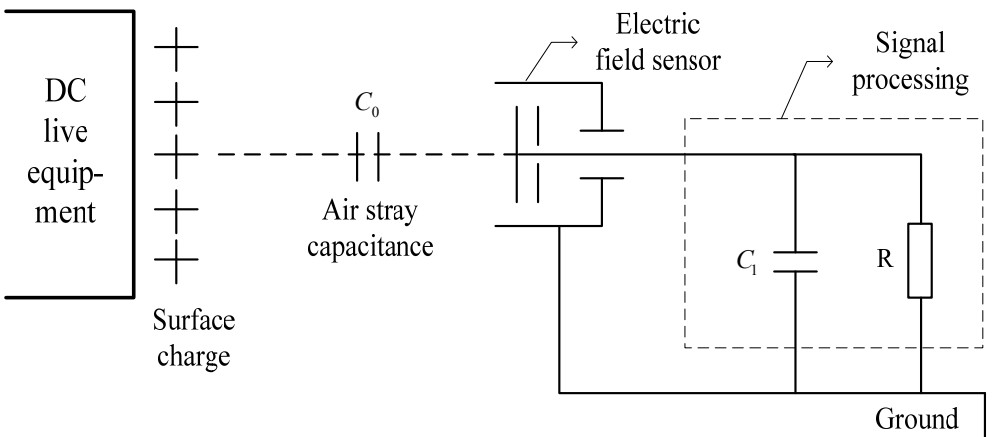

**Figure 1.** Schematic of non-contact measurement method.

In Figure 1, $C_0$ represents the air stray capacitance value, which can be ignored generally. $C_1$ and $R$ represent signal processing circuits at the back-end of the sensor.

In the process of electric field measurement, a small amount of charge flows from the charged device into the sensor, or leaks out, which will cause a large measurement error. Moreover, the structure of the sensor itself will have a distortion effect on the original electric field distribution. The sensor is required to have a small coupling capacitance and a very high input impedance to reduce the measurement error [11].

*2.2. Sensor Measurement Principle*

The electric field sensor uses induced charge to measure the electric field strength. Since the high-voltage DC electric field is a quasi-static electric field, then that sensor measures the electric field according to Gauss's theorem [12]. That is, the flux of the electric field strength vector through the closed surface is equal to the ratio of the free charge algebra and the dielectric constant enclosed by the closed surface as Equation (1).

$$\oint_s \vec{E} \cdot d_{\vec{s}} = \frac{1}{\varepsilon_0} \cdot \int_V \rho \cdot d_V = \frac{1}{\varepsilon_0} \cdot Q \tag{1}$$

It can be known from Equation (1), that when the sensor is exposed to the electric field $\vec{E}$, the electric flux of the electric field line passing through the effective area $S$ of the induction electrode is equal to the ratio of the free charge $Q$ accumulated on the induction electrode area $S$ to the dielectric constant.

The electric motor of the electric field sensor drives the shield electrode to rotate. This is so that the induction area of the induction electrode exposed to the electric field $\vec{E}$ changes periodically. To maintain the same potential as the ground, a certain amount of charge is accumulated on the sensing electrode. When the motor drives the shield electrode to rotate, the sensing electrode is in three states of "exposure-shielding-shielding" [13–15], and the induced charge $Q(t)$ on the sensing electrode changes periodically, and an alternating induction will occur on the sensing electrode current signal $i(t)$ as Equation (2).

$$i(t) = \frac{dQ(t)}{dt} = \varepsilon_0 E \cdot \frac{dS(t)}{dt} \tag{2}$$

where, $Q(t)$ is amount of induced charge, $C$; $S(t)$ is the area of the sensing electrode, m$^2$.

It can be known from the equation (2), the magnitude of the electric field strength $E$ is proportional to the induced current signal $i(t)$. The induced current signal is processed through a series of signals to obtain a measurable DC voltage signal. The electric field strength can be calculated based on the voltage values which are linear.

## 3. Development of Electric Field Sensor

Developing a non-contact high-voltage DC electric field measurement method which can help to safely alarm DC high voltages. In addition, the focus is on the development of DC electric field sensors. Effectively ensuring accurate non-contact measurement and the safety alarming of high-voltage DC electric fields is essential. The DC electric field sensor first uses a mechanical induction device to sense the electric field signal, and then the signal is conditioned by the internal circuit. The overall structure can be divided into mechanical and electrical parts. The mechanical part is mainly about the electric field induction device and motor and housing. The circuit part is mainly about the signal conditioning circuit and motor speed control circuit. The overall structure of the electric field sensor is shown in Figure 2.

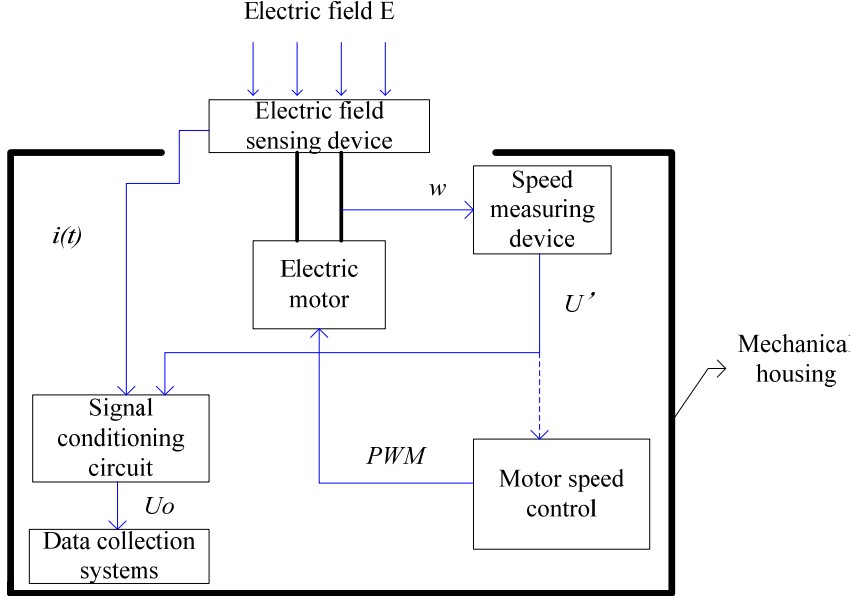

**Figure 2.** Overall structure of electric field sensor.

The shaft of the motor drives the shield electrode to rotate. So that the area of the sensing electrode exposed to the electric field changes. Equipped with a photoelectric switch, which can measure the motor speed $\omega$ as the feedback signal $U'$ of the motor closed-loop control, a circuit board is installed below the motor, which includes an MCU and a signal conditioning circuit. The motor speed is controlled by PWM. The signal conditioning circuit processes the current $i(t)$ generated by the induction device and finally obtains the voltage $U_O$.

### 3.1. Sensor Sensing Device

In order to facilitate the structural design and processing of the sensor sensing device, the shield electrode and the sensing electrode blade adopt a fan-shaped opening method. The actual parameters of the number of opening lobes $n$, the inner diameter $r$ of the fan, the outer diameter $R$ of the fan and the vertical distance $d$ between the two electrodes of the shield electrode and the sensing electrode, are shown in Table 1.

**Table 1.** Shielding electrode and induction electrode blade processing parameters.

| Number of Flaps *n* | inside Diameter *r* | Outer Diameter *R* | Blade Pitch *d* |
|:---:|:---:|:---:|:---:|
| 6, 12 | 15 mm | 25 mm | 3 mm |

The shield electrode is made of copper sheet with a thickness of 1 mm, and the number of opening lobes of the shield electrode blade is $n = 6$, which is grounded to the metal shaft of the three-phase brush-less DC motor. The sensing electrode is made by PCB processing technology, and the process error is only ±0.1 mm. The number of perforated lobes of the sensing electrode blades is $n = 12$. It adopts a 12-division blade structure. Six spaced sectors form a group, and the two signals form a differential signal output. The two sets of sectors of the sensing electrode are connected to the pads with copper wires, respectively. In addition, the signal lines drawn from the pads are then connected to the signal conditioning circuit. The actual processing of the induction plate and shield plate of the electric field sensor induction device is shown in Figure 3.

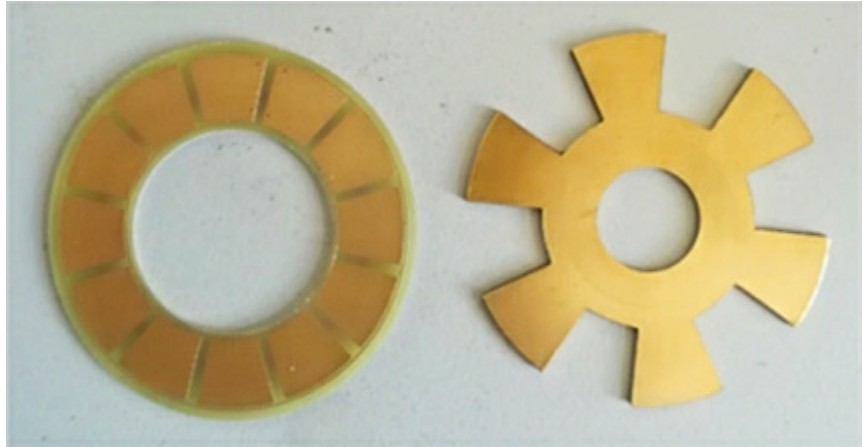

**Figure 3.** Physical drawing of the induction and shield plates.

According to the above, the expression of the area of the sensing electrode blade over time can be obtained as Equation (3).

$$S(t) = \begin{cases} \frac{1}{2} \cdot 12\omega t \cdot (R^2 - r^2), & 0 \le t < \frac{T}{2} \\ (\frac{\pi}{2} - \frac{1}{2} \cdot \omega t) \cdot (R^2 - r^2), & \frac{T}{2} \le t < T \end{cases} \tag{3}$$

where, $T$ is the period of the area change. And $\omega$ is the angular velocity of the motor rotation. Substituting Equation (3) into (2) to get the induced current signal as Equation (4).

$$i(t) = \begin{cases} \frac{1}{2} \cdot 12\varepsilon_0\omega \cdot (R^2 - r^2)E, & 0 \le t < \frac{T}{2} \\ -\frac{1}{2} \cdot \varepsilon_0\omega \cdot (R^2 - r^2)E, & \frac{T}{2} \le t < T \end{cases} \tag{4}$$

When the sensing electrode is completely exposed to the electric field, the charge sensing area is the largest, and the sensing current on the sensing electrode is also the largest. When the sensing electrode is completely shielded by the shield electrode, the charge sensing area is zero. Meanwhile, the sensing electrode releases the induced charge from the previous cycle to prepare for the next cycle.

### 3.2. Sensor Signal Processing Circuit

The electric field sensor signal processing hardware circuit mainly includes differential amplification, peak detection, low-pass filter, motor drive and electric field polarity discrimination. The sensor signal processing circuit is composed as shown in Figure 4.

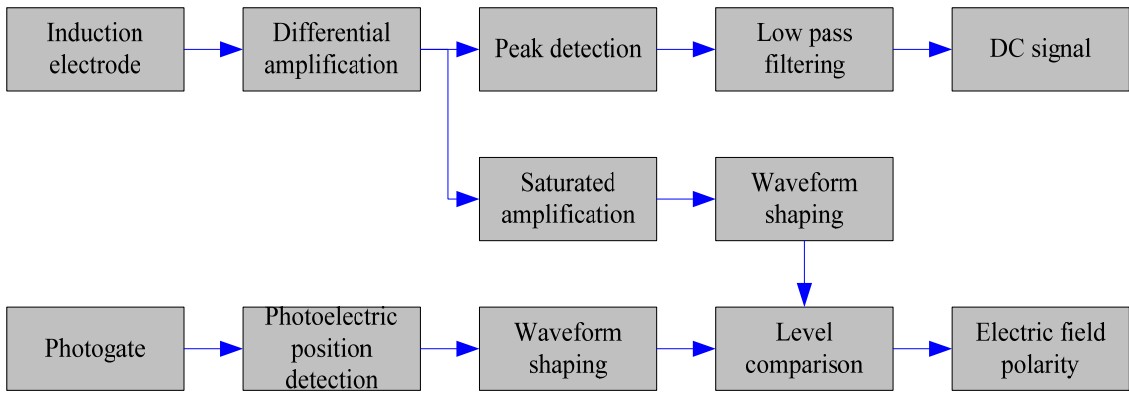

**Figure 4.** Sensor signal processing circuit.

Here we have the explanation of differential circuit function.

After the motor speed stabilizes, two induction current signals with equal amplitude and 180° phase difference will be generated on the induction electrode. Take this differential signal as the sensor output signal to suppress the common mode signal interference and improve the measurement accuracy [16]. The weak induced current signal is transmitted to the AD620 amplifier chip, so that the differential form of the alternating induced current signal is converted into a voltage signal.

Here we have the explanation of rectification and peak detection circuit function.

Because the voltage signal after the differential amplification circuit is positive and negative bipolar. It needs to be changed to positive polarity by full-wave rectification, which is convenient for microprocessor ADC acquisition. In order to facilitate software processing, the peak value of the signal is used to output the DC voltage of the signal. Measuring and displaying the voltage signal value to provide actual measurement parameters for the design of the early warning algorithm for high-voltage operation safety.

Here we have the explanation of motor drive circuit function.

A three-phase DC brush-less motor is selected and the MCP8063 driver chip is used. The motor speed is controlled by pulse width modulation (PWM). The minimum driving voltage is 2.5 V, and the minimum working current is less than 5 mA. The MCP8063 driver includes a lock protection mode that turns off the output current when the motor is in a locked state. Features such as motor over-current limit and thermal shutdown protection can improve the reliability and stability of the motor system [17,18].

It can be known from Figure 5 that the motor speed is faster and the amplitude of the output signal of the sensor is larger. But in the actual engineering applications, the excessively fast motor speed will increase the power consumption of the entire sensor and shorten the sensor service life.

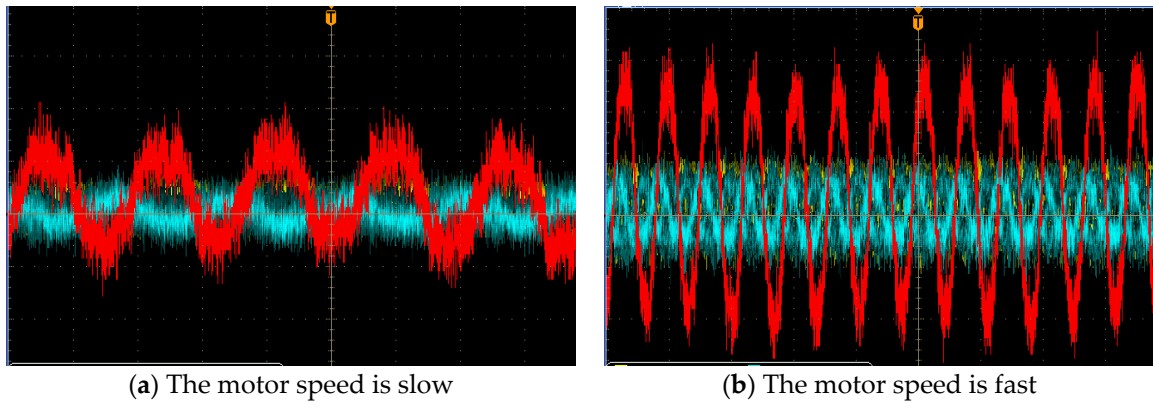

(**a**) The motor speed is slow          (**b**) The motor speed is fast

**Figure 5.** Waveform of sensor output signal changing with motor speed.

Too slow motor speed will also reduce the sensitivity of the sensor and affect the accuracy of the electric field measurement. In order to study and quantify the effect of changing the motor speed on static index parameters, such as the sensor sensitivity coefficient and range [13], this paper performed detailed experiments and data analysis on the sensor static index calibration experiment.

Here we have the explanation of electric field polarity discrimination circuit function.

The level comparison circuit can judge the positive and negative polarity of the measured high-voltage DC electric field. It can be known from Figure 6 that if the motor drive signal (yellow) is in phase with the sensor output signal waveform (blue), that means the measured high-voltage DC electric field polarity is positive. If the waveform of the above two phases are inverted, that means the measured high-voltage DC electric field polarity is negative.

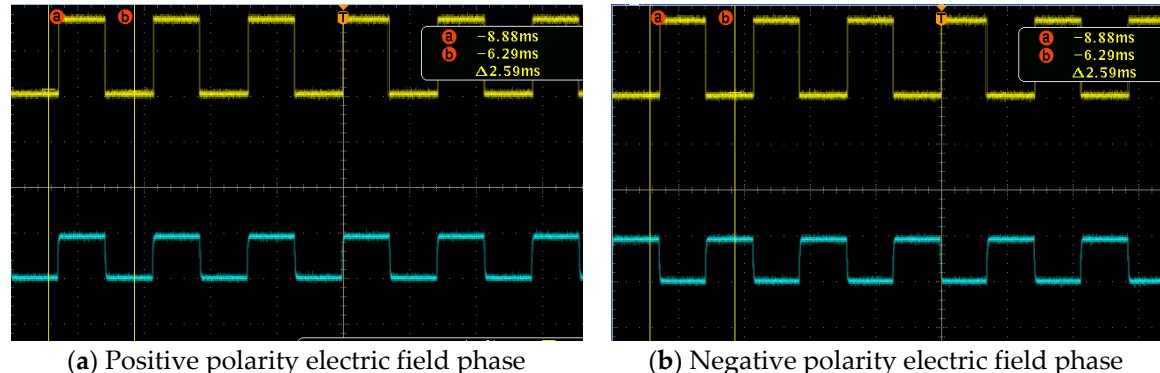

(**a**) Positive polarity electric field phase　　　　　(**b**) Negative polarity electric field phase

**Figure 6.** Discrimination positive and negative polarity of electric field.

In the detection of the sensor output signal phase and the motor rotation drive signal phase, the Schmitt trigger is used. It can play a waveform shaping function. In addition, it can be inhibiting the waveform from generating jitters or glitches to turn the signal into a standard square wave signal [19].

### 3.3. Comparative Analysis of Sensors

In order to verify the consistency and reliability of the above-mentioned electric field sensors, this paper studied the theoretical simulation calculations and calibration experiments. As well as this, the paper conducted a comparative analysis of each test result.

### 3.3.1. Sensor Theoretical Calculation

It is assumed that the amount of induced charges when the sensing electrode is completely blocked by the shield electrode is $Q_1$, and the amount of induced charges when the sensing electrode is completely exposed to the electric field is $Q_2$. From the principle of the differential signal we know that $Q_1 = Q_2$, and the phase difference between the two is 180 degrees. When the sensing electrode changes from a fully shielded state to a fully exposed state, the amount of change in the induced charge on the sensing electrode is as in Equation (5).

$$q = 2Q = 2\varepsilon_0 |E| \cdot S \tag{5}$$

According to the Equation (5) and the sector area equation, the change amount of the induced charge on the sensing electrode under different electric field strengths can be obtained. Then we can calculate the value of the induced current signal based on the theory to obtain the DC voltage signal value of the sensor output. According to Equation (2), this is as:

$$i(t) = \frac{dQ(t)}{dt} = \frac{\Delta q}{T/2} = 2f \cdot q \tag{6}$$

where $f$ is the motor drive frequency, Hz; $\Delta q$ is the amount of charge change on the sensing electrode in a cycle, C.

From the amplification factor of the input and output signals of the various stages of the signal conditioning circuit at the back of the electric field sensor, we see the relationship between the induced current signal output by the electric field sensor sensing device. Along with this, the DC voltage signal output by the signal conditioning circuit is calculated as Equation (7).

$$U_O = 5.714 \cdot 10^7 \cdot i(t) \tag{7}$$

In summary, it is possible to calculate the charge change amount, induced current value and sensor output DC voltage value of the sensor plate under different electric field intensities as shown in Table 2.

**Table 2.** Calculation results of various changes under different electric field strengths.

| E (kV/m) | q (pC) | i(t) (nA) | $U_O$ (V) |
|---|---|---|---|
| 2.5 | 2.78 | 1.36 | 0.09 |
| 10 | 11.1 | 5.42 | 0.31 |
| 17.5 | 19.4 | 9.47 | 0.54 |
| 25 | 27.7 | 13.5 | 0.77 |
| 32.5 | 36.1 | 17.6 | 1.01 |
| 40 | 44.3 | 21.6 | 1.23 |
| 47.5 | 52.7 | 25.7 | 1.46 |
| 55 | 61.1 | 29.8 | 1.71 |
| 62.5 | 69.4 | 33.9 | 1.94 |

It can be known from Table 2 that the amount of change in the induced charge on the sensing electrode increases with the increase of the applied electric field intensity, and there is a linear relationship between the two. The DC voltage value $U_O$ output by the sensor also increases with the increase of the applied electric field strength $E$, which is linearly proportional to the magnitude $i(t)$ of the induced current.

### 3.3.2. Sensor Calibration Experiment

The electric field sensor is the core component of the high-voltage DC safety early warning device. Before using, a strict performance test must be performed according to the test rules to determine the characteristic parameters of the sensor. The effects of the calibration experiment device generating standard DC electric field accuracy, the measurement electric field range and the calibration method on the electric field sensor calibration test parameters, will directly affect the accuracy and reliability of the electric field sensor in actual using [20,21].

Calibration of the sensor is a prerequisite for accurate measurement of the electric field. Calibration of the electric field sensor is performed in a uniform electric field generated by two parallel plates forming a capacitor. It can be known from Figure 7 that the principle of sensor calibration is as:

$$E = \frac{U}{d}$$

(8)

where $E$ is the electric field strength between the two plates; $U$ is the DC voltage between the two plates; $d$ is the distance between the two plates.

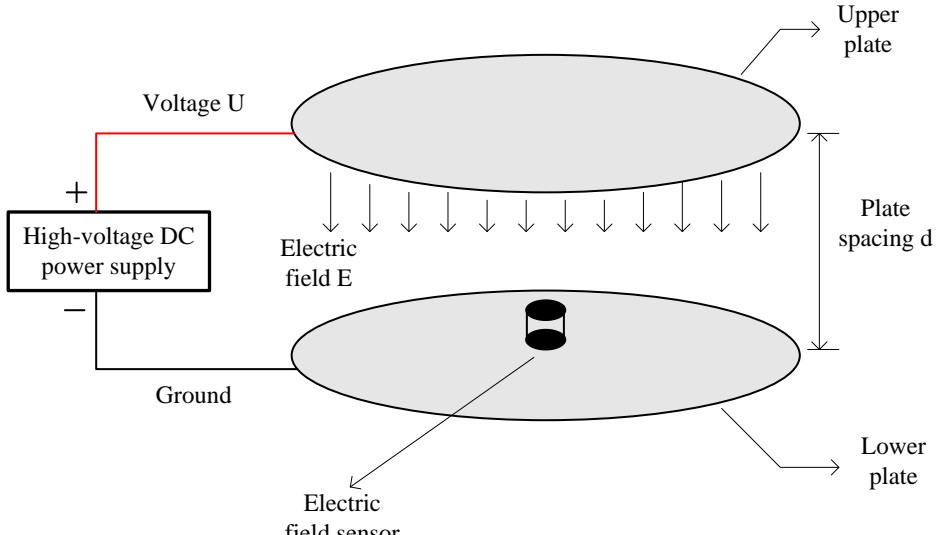

**Figure 7.** Sensor calibration experimental device model.

The DC voltage $U$ is applied to the upper plate of the parallel plate capacitor and the lower plate is grounded. An electric field $E$ is generated between two plates. The ratio of the area of the capacitor to the distance between the plates should be as large as possible to reduce the effect of edge effects [22]. Table 3 shows the material and size of the parallel plate capacitor.

**Table 3.** Sensor calibration experimental device parameters.

| Plate Material | Plate Shape | Plate Thickness | Plate Diameter | Plate Spacing | DC Voltage Range |
|---|---|---|---|---|---|
| Stainless steel | Round | 1 mm | 1 m | 20 cm | 0~30 kV |

The upper plate of the parallel plate capacitor is connected to a high-precision high-voltage DC power supply which can output 0~30 kV. The DC power supply of ripple time drift accuracy is 0.1%/h to meet the requirements of calibrated sensors [23].

This is after turning on the electric field sensor, placing the sensor in the middle position of the DC electric field calibration platform and then connecting the high voltage DC power supply. The electric field sensor calibration experiment test site is shown in Figure 8.

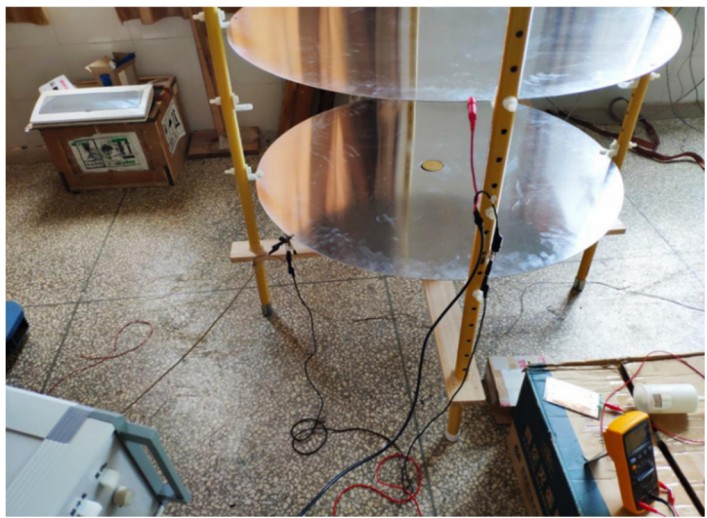

**Figure 8.** Sensor calibration experiment test site.

The entire experimental device needs to be reliably grounded to prevent electric shocking and charge accumulation discharging. After the DC voltage is greater than 30 kV, it is necessary to pay attention to whether the experimental device has a discharge phenomenon which would make the measurement inaccurate. The experiment results are shown in Figure 9.

In Figure 9, curve a is the curve of the measured value of the sensor as a function of the electric field strength, and curve b is the curve of the theoretical calculation value of the sensor fitted by the least square method as a function of the electric field strength. Overall, the two curves have good consistency and linearity. There is a significant deviation between the two data points near the sensor output DC voltage $U_O = 1.5$ V due to reading errors.

According to the calibration experiment test data, the maximum parameters, self-resolution, linearity and measurement errors of the sensor are calculated, as shown in Table 4.

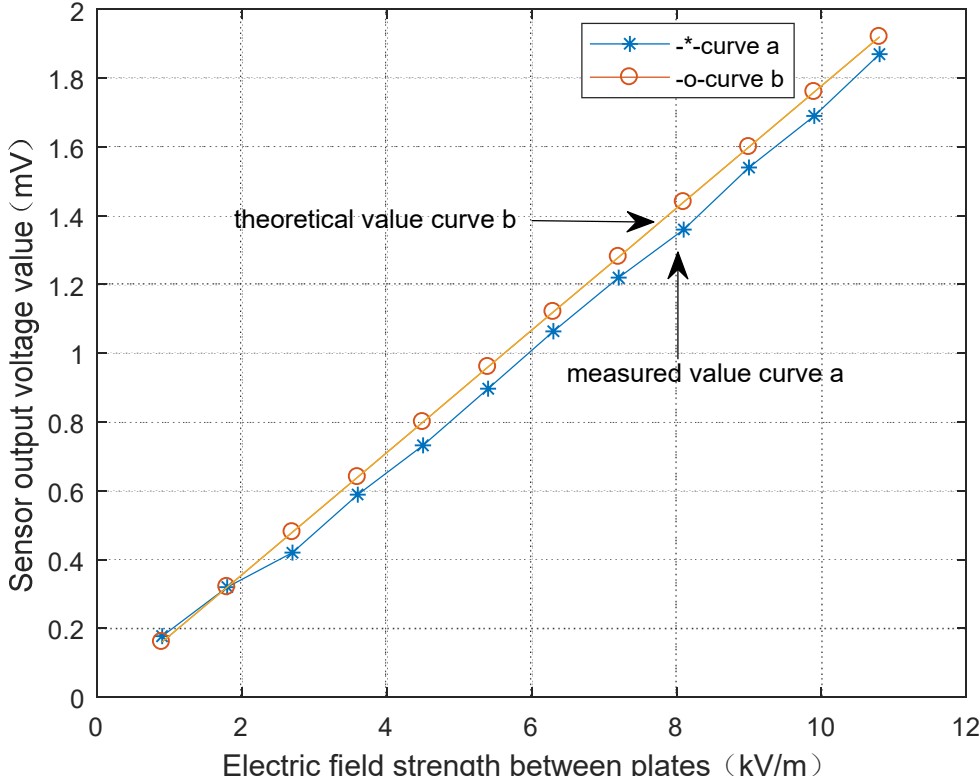

**Figure 9.** Comparison curve of measured value and theoretical value.

**Table 4.** Sensor experimental performance parameters.

| Range | Sensitivity | Precision | Linearity | Measurement Error |
|---|---|---|---|---|
| 0~±114 kV/m | 30.91 | 0.05 kV/m | 2.08% | 3.01% |

According to the sensor characteristic parameters in Table 4, it can be seen that the measured high-voltage DC electric field measurement range is 0~±80 kV/m, and the sensor measurement error is less than 5%. This is in order to make the electric field sensor measurement more accurate, which needs to be further improved and perfected.

According to the above, changing the speed of the motor will affect the sensitivity and range of the sensor, and further experimental tests to quantify and analyze the degree of impact. In the experiment, the mobile phone and the electric field sensor are used to establish a Bluetooth wireless communication connection. The mobile terminal APP sends instructions to the sensor to change the frequency of the motor drive signal, thereby changing the motor speed. For the convenience of observation and analysis, this article takes nine instructions, and each instruction corresponds to a duty cycle of the PWM wave, which ranges from R1-20% to R9-100%. Based on the measured value of the sensor and subtracting the ambient noise, the influence of the motor speed on the sensor sensitivity is shown in Figure 10.

According to the experimental results, it can be known that the larger the motor speed, the greater the sensor sensitivity, which values have a linear relationship. Increasing the sensitivity of the sensor enables higher measurement accuracy. However, the higher the sensitivity makes the measurement range to be narrower and the stability to be worse [24].

Similarly, the motor speed also has a linear relationship with the sensor range, and the smaller motor drive signal and the larger sensor range. A comparative analysis of R9-100% and R4-50% shows that when the motor speed is reduced by 50%, which causes the range is doubled, then the sensitivity is halved.

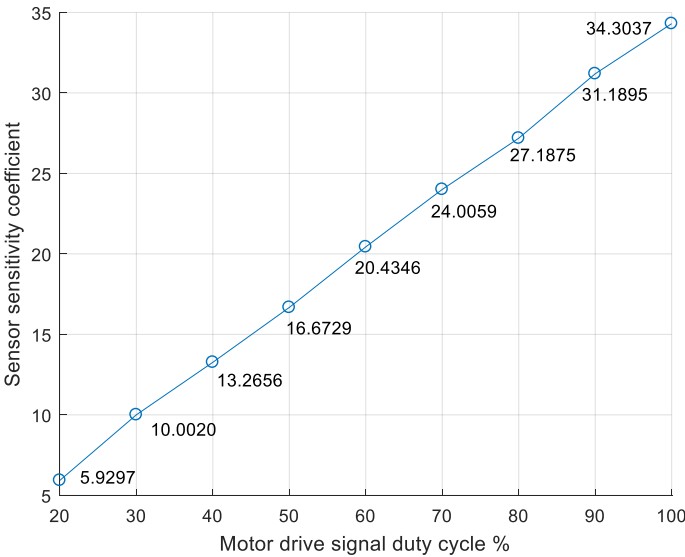

**Figure 10.** Effect of motor speed on sensor sensitivity.

## 4. Development of Early Warning Devices

### 4.1. Security Early Warning Implementation Plan

The early warning device system consists of an electric field sensor, a microprocessor and peripheral circuit systems, Bluetooth communication, a matching safety early warning mobile phone APP and a remote database. The implementation scheme of HVDC electric field adaptive safety early warning technology is shown in Figure 11.

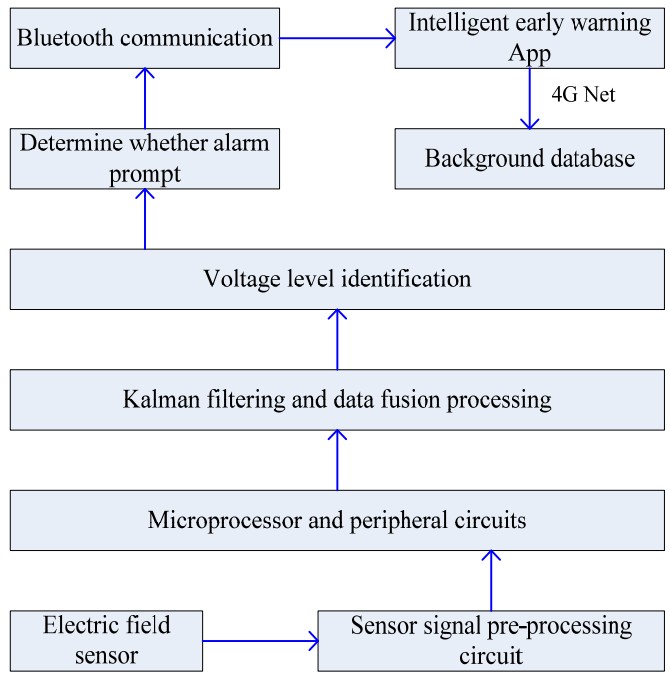

**Figure 11.** Security early warning technology implementation plan.

The electric field sensor developed in this paper is used to measure the electric field strength around the DC-charged equipment, and the electric field signal measured by the sensor is collected by the ADC to the microprocessor of the early warning device. The microprocessor compares the measured value of the electric field with the set safety threshold. If the safety threshold is exceeded, the early

warning device issues an alarm prompt and transmits the data to the mobile phone through Bluetooth wireless communication. It can be seen directly that the real-time waveform of the measurement electric field and the alarm prompt symbol on the safety alert the APP interface.

### 4.2. Adaptive Security Early Warning Algorithm

For different voltage levels of high-voltage DC live equipment, such as ±500 kV and ±800 kV, the early warning device can identify different voltage levels of live equipment, and the electric field sensor adjusts the range and sensitivity dynamically, which bases on the voltage level recognized by the microprocessor to meet the measurement requirements. The voltage level identification and related logic processing are performed by obtaining electric field sensing data and performing filtering processing and data fusion. In addition, a corresponding voltage level identification algorithm needs to be designed.

The electric field strength value increases exponentially when approaching high-voltage DC charged equipment. The gradient of the electric field strength changes to the maximum when the safety distance is reached [25–27]. The safety distance is different for different voltage levels. That is, the position of the electric field change gradient is distinguished. The electric field strength gradient $\nabla E$ can be expressed by the change of the electric field $E$ with time.

$$\nabla E = \frac{dE}{dt} \tag{9}$$

In this paper, ±500 kV and ±800 kV HVDC transmission lines are taken as examples to analyze the calculation of their spatial electric field strength and the variation of the electric field strength gradient. The electric field strength is generated by the electric charge of the transmission line [23–25]. The calculation of the electric field strength is divided into two steps, as:

(1) Calculate the equivalent charge on a unit length DC transmission line using the Maxwell coefficient method;
(2) Calculate the strength of the electric field generated by the electric charge according to the mirror image method.

Assuming that the HVDC transmission line is an infinitely long straight wire and parallel to the ground. The equivalent charge of the transmission line is solved by the mirror method as shown in Figure 12.

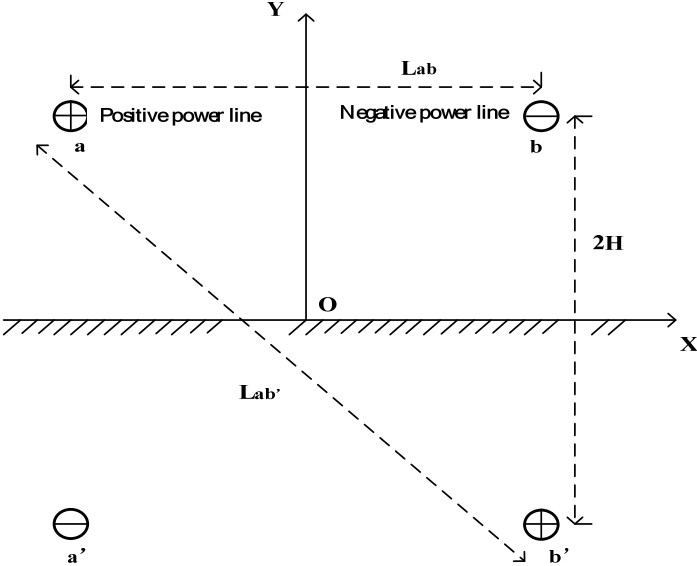

**Figure 12.** Mirror Method for Solving the Equivalent Charge of Transmission Lines.

Solving the equivalent charge on the transmission line which bases on the following matrix equation.

$$V = \lambda \cdot Q \tag{10}$$

where, $V$ is the operating voltage of the transmission line to ground, kV; $\lambda$ is the Maxwell potential coefficient, m/F; $Q$ is the equivalent charge of the transmission line, C/m.

According to equation (11) and (12), the Maxwell potential coefficient can be obtained.

$$\lambda_{a \to a} = \frac{1}{2\pi\varepsilon_0} \cdot \ln\left(\frac{2H}{r}\right) \tag{11}$$

$$\lambda_{a \to b} = \frac{1}{2\pi\varepsilon_0} \cdot \ln\left(\frac{L'_{ab}}{L_{ab}}\right) \tag{12}$$

where, $\varepsilon_0$ is the dielectric constant of vacuum; $H$ is the height of transmission line a from the ground, m; $r$ is the equivalent radius of transmission line a, cm. $L_{ab}$ is the distance between the positive polarity power line a and the negative polarity power line b, m. $L'_{ab}$ is the distance from the positive polarity power line a to the mirror negative power line b', m.

The equivalent charge of the transmission line and its mirrored transmission line can be obtained according to Equation (10), and the magnitude and direction of the electric field can be calculated. The electric field component in the horizontal X direction is equation (13).

$$E_x = \sum_{a=1}^{n} \frac{Q_a}{2\pi\varepsilon_0} \cdot \frac{x_a}{x^2 + (y - y_a)^2} \tag{13}$$

where, $Q_a$ is the equivalent charge of power line a; $(x_a, y_a)$ is the coordinate of power line a; the total number of power lines and mirror power lines is $n = 4$.

Similarly, the electric field component in the vertical Y direction as Equation (14).

$$E_y = \sum_{a=1}^{n} \frac{Q_a}{2\pi\varepsilon_0} \cdot \frac{y - y_a}{x^2 + (y - y_a)^2} \tag{14}$$

The magnitude and direction of the electric field strength along the transmission line can be obtained as in Equation (15):

$$E = \sqrt{E_x^2 + E_y^2}, \tan\theta = \frac{E_y}{E_x} \tag{15}$$

The electric field strength gradient $\nabla E$ can be calculated from the above:

$$\nabla E = \frac{d\sqrt{E_x^2 + E_y^2}}{dt} \tag{16}$$

In summary, the method for calculating the electric field variation gradient of DC transmission lines is theoretically derived. It lays the theoretical foundation for the electric field distribution simulation calculation and electric field gradient change analysis of DC transmission lines with different voltage levels.

A two-dimensional calculation model was established according to the actual DC transmission line, and the COMSOL software was used to simulate the spatial electric field distribution of the ±500 kV and ±800 kV DC transmission lines. The two-dimensional calculation model simplifies the actual scene and ignores the influence of metal fittings such as transmission towers and insulator strings [28]. It is assumed that the transmission line is a wireless, long, straight wire parallel to the horizontal ground. The vertical height of the line is the vertical distance from the ground to the lowest position of the arc. The two-dimensional calculation model is shown in Figure 13.

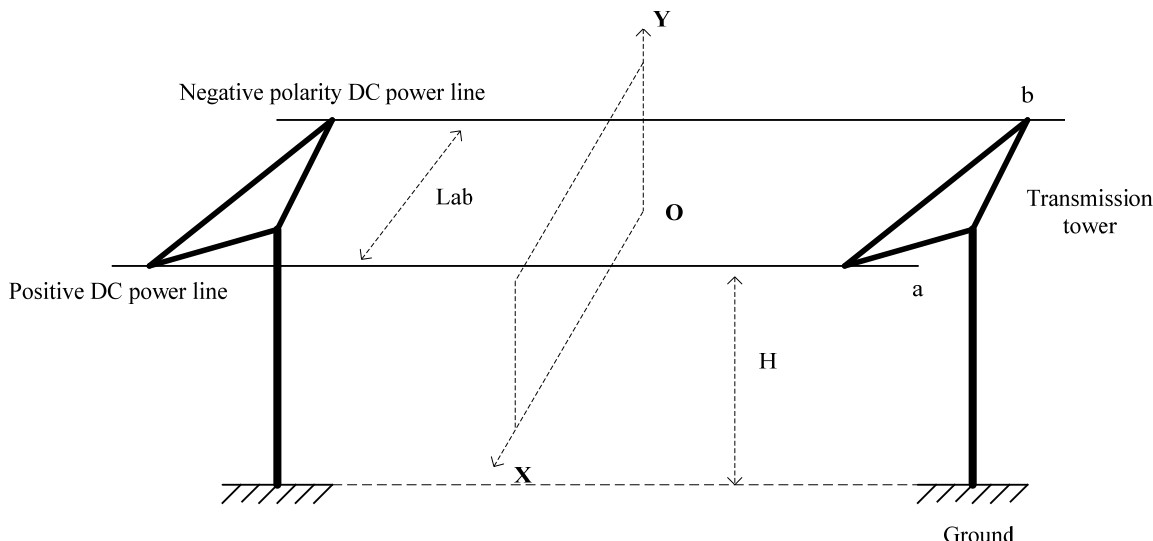

**Figure 13.** Two-dimensional calculation model of the high-voltage, direct current (HVDC) transmission line.

This paper analyzed the ±800 kV DC transmission line. According to its actual operating parameters, the height of the transmission line from the ground is $H$ = 18 m and the distance between the positive and negative transmission lines is $Lab$ = 22 m. The XOY plane perpendicular to the transmission line is used as the calculation surface. The spatial potential contour of the ±800 kV DC transmission line is shown in Figure 14.

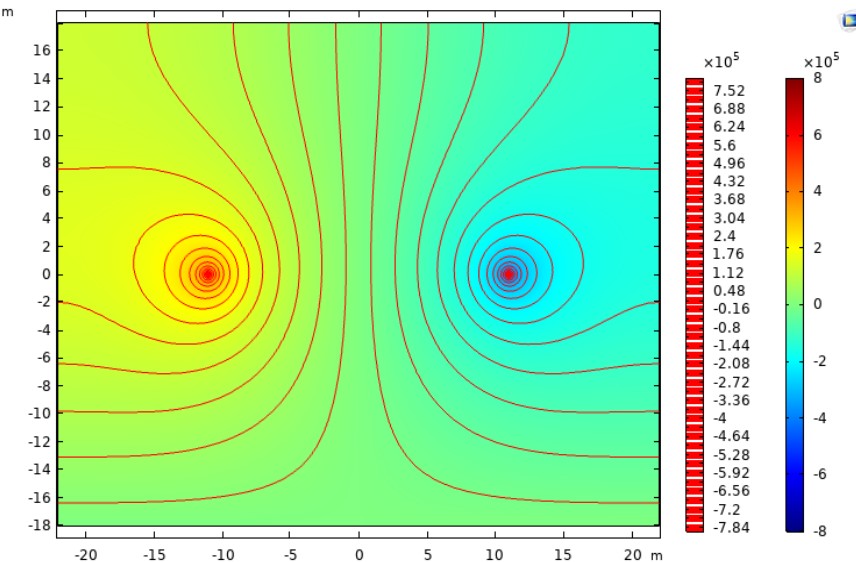

**Figure 14.** Space potential contour map of ±800 kV DC transmission line.

It can be seen from Figure 14 that the spatial potential of the ±800 kV DC transmission line is symmetrically distributed and the electric field distribution is also symmetrically distributed. The left side is a positive transmission line and the right is a negative transmission line.

Because the simulation is based on actual operating parameters, the distance between the positive and negative transmission lines is relatively long. This was in order to observe the changes of the electric field intensity in the space around the DC transmission line more closely. The cloud diagram of the distribution of the electric field in the space around the positive transmission line is taken as shown in the Figure 15.

It can be seen from Figure 15 that the center position is a direct current transmission line and the electric field strength is attenuated from the inside to the outside, and the change of the electric field

can be clearly seen. The electric fields at the two positions marked by the purple wire-frames in the picture are significantly distorted. The electric field distribution is distorted due to the simulation calculation of the grid corners.

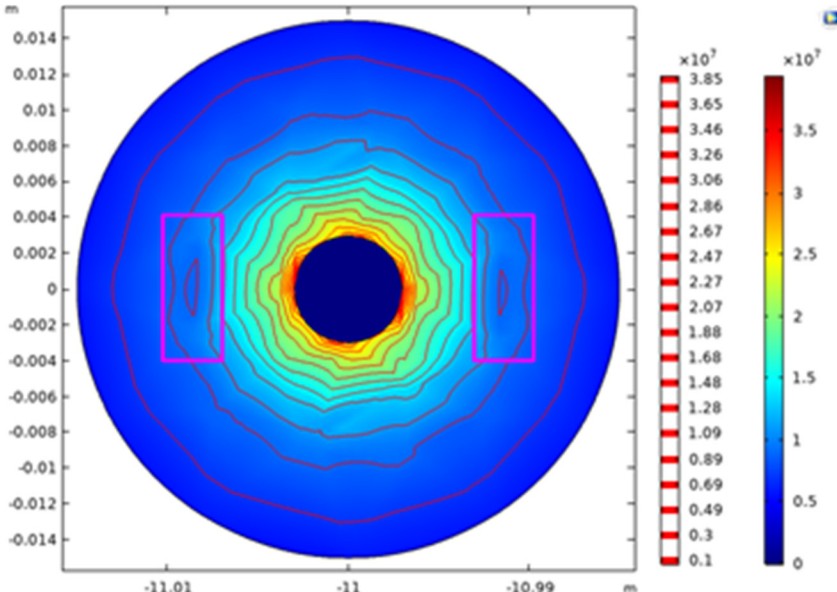

**Figure 15.** Cloud diagram of electric field distribution of a positive transmission line.

The vertical electric field distributions of ±500 kV and ±800 kV bipolar DC transmission lines calculated by COMSOL software are shown in Figure 16.

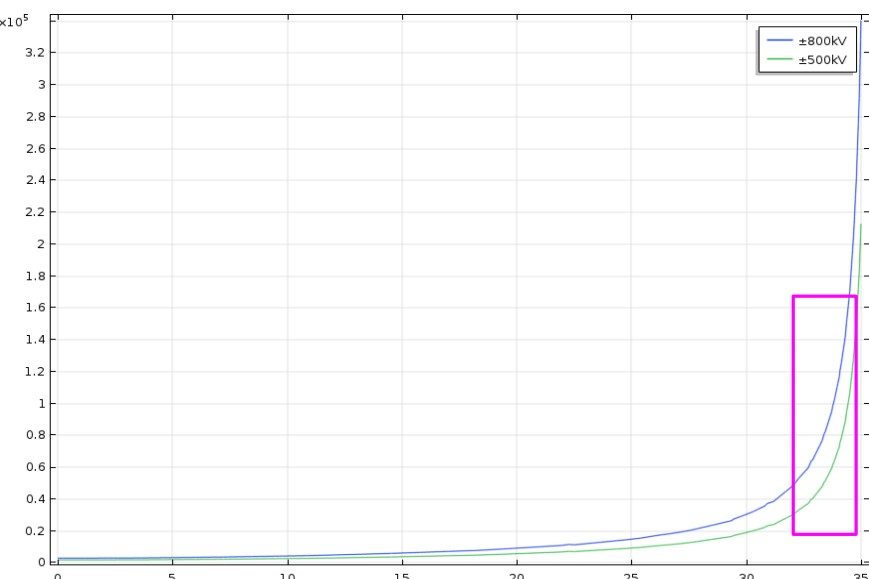

**Figure 16.** Electric field distribution curve in vertical direction of transmission line.

The horizontal electric field distributions of ±500 kV and ±800 kV bipolar DC transmission lines calculated by COMSOL software are shown in Figure 17.

As it can be seen from Figures 16 and 17 that the green curve represents a ±500 kV electric field distribution curve and the blue curve represents a ±800 kV electric field distribution curve. Meanwhile, with vertical height or horizontal distance, the voltage level is higher and the electric field strength is greater. This trend becomes more distinct as it approaches the high-voltage source until it is near the safe distance which can be used as a reference for safe distance warning.

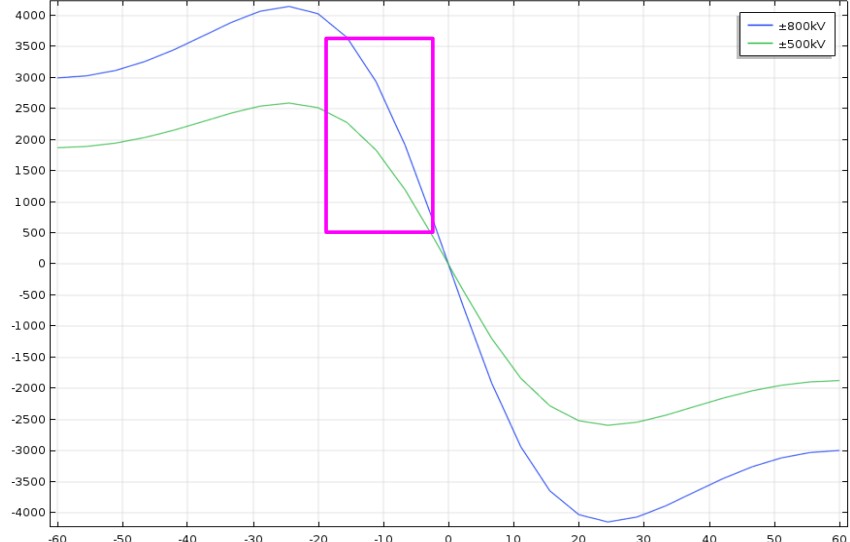

**Figure 17.** Electric field distribution curve in horizontal direction of transmission line.

Generally, the value of the electric field strength near the safety distance is selected and obtained through multiple measurements and statistics. Calculating the electric field intensity gradient value or the unit distance change rate $\nabla E$ of the electric field. Different voltage levels of DC charged equipment are distinguished by gradient values of different electric field strengths. However, this algorithm is susceptible to the speed of human movement, and the influences can be eliminated by Karlman filtering [29,30]. Based on the above principles, the design flow of the adaptive security early warning algorithm is shown in Figure 18.

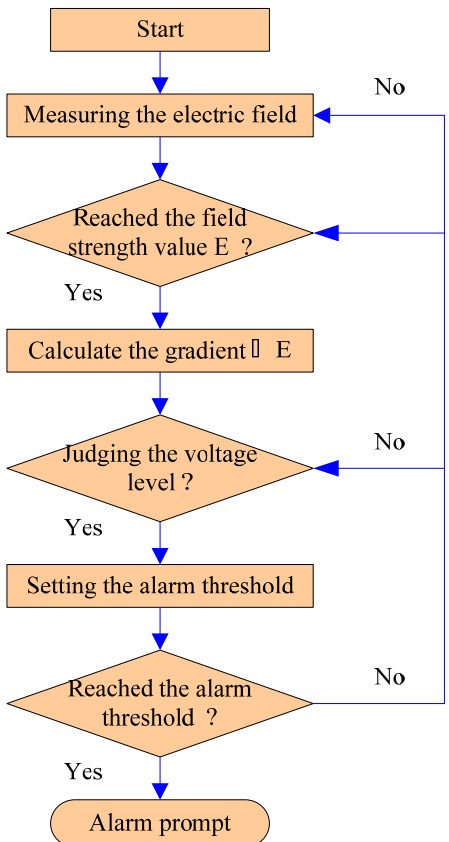

**Figure 18.** Adaptive Security Early Warning Algorithm.

If in the DC converter station, the early warning electric field strength can be automatically calculated according to the ground reference electric field strength, and the state judgment can be made based on the height change information. It is not necessary to distinguish the voltage level to achieve accurate early warning.

*4.3. Early Warning Device Field Test*

In order to verify the actual reliability of the high-voltage DC electric field adaptive safety early warning device. An onsite working condition test was selected in a ±500 kV DC converter station. The early warning device test site is shown in Figure 19.

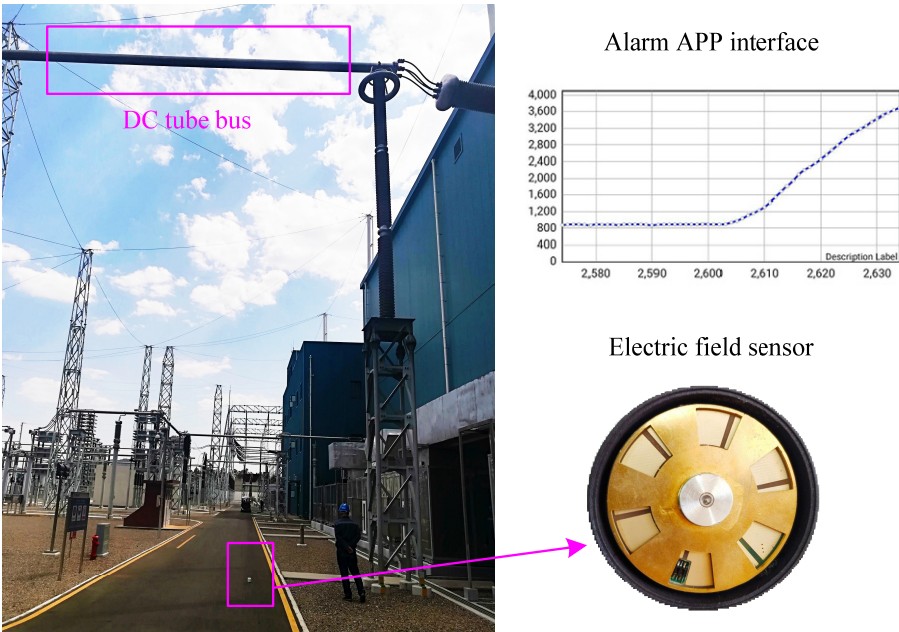

**Figure 19.** Field test of ±500 kV DC converter station.

At the +500 kV pole-one outlet position of the DC converter station, the early-warning device measured the space electric field strength at that location to be 5.8 kV/m, and the space electric field strength at the location was about 5.5 kV/m through simulation calculations. The simulation values and actual measurements. The values differ by 0.3 kV/m. The measured electric field strength is basically consistent with the simulation results of the field conditions which indicates that the electric field sensor used in the early warning device has good reliability. It can provide a reliable electric field value for the early warning device and ensure accurate alarm prompts.

The warning device will give an alarm when the safety warning device is $d0 = 7.1$ m from the ±500 kV live equipment. This is the minimum safe distance $d1 = 6.8$ m corresponding to the ±500 kV DC live equipment stipulated in the "Power Safe Working Rules and Power Lines" in Table 5. The difference between the two safety distances is 0.3 m, which is used as the safety distance redundancy [30]. The early warning requirements have been met, and there are no omissions and false alarms which indicating that the early warning device meets the requirements.

**Table 5.** Minimum safety distance between workers and live equipment.

| Voltage Level/kV | Safety Distance/m |
| :---: | :---: |
| ±500 | 6.8 |
| ±660 | 9.0 |
| ±800 | 10.1 |

In order to further verify the reliability of this high-voltage DC electric field adaptive safety early warning device. Four simulation of early warning test experiments of +500 kV, −500 kV, +800 kV and −800 kV were performed, and the high-voltage simulation test platform is shown in Figure 20.

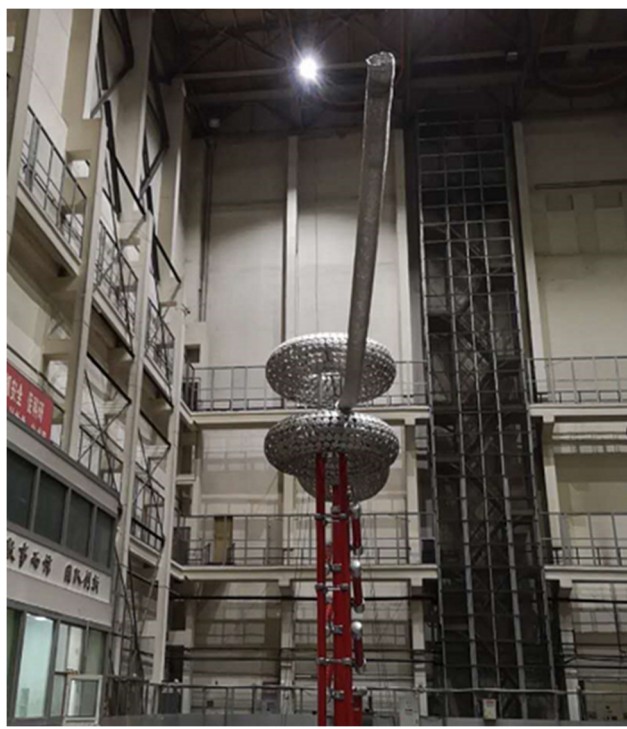

**Figure 20.** High-voltage simulation test platform.

Using the early-warning device developed in this article to simulate early-warning tests on four different voltage levels. It can accurately reflect the changing trend of the electric field in the space around the DC charged equipment. The test results are shown in Table 6 as:

**Table 6.** Results of safety warning tests at different voltage levels.

| Voltage Level/kV | Safety Distance/m |
| --- | --- |
| +500 | 7.3 |
| −500 | 6.9 |
| +800 | 11.2 |
| −800 | 10.5 |

Comparing Tables 5 and 6, it can be known that the safety distance of the early warning device for the warning test of ±500 kV voltage level is greater than 6.8 m, and the safety distance for the warning test of ±800 kV voltage level is greater than 10.1 m. In addition, in the early warning test of the −500 kV and +800 kV voltage levels, the safety alarm distance is very close to the value specified in Table 5, which shows that the early warning device accurately reports the alarm, and there are no false alarms [31].

## 5. Discussion and Conclusions

This paper proposed an adaptive security early warning technology solution based on non-contact electric field measurement. The entire safety early warning system device consists of an electric field measurement sensor and a mobile phone terminal early warning APP. The early warning system uses a voltage level identification algorithm to identify the voltage level of the high-voltage DC live equipment. According to the measurement requirements of different voltage levels, the system microprocessor

sends instructions to control the motor speed to change the measurement range and the sensitivity of the sensor, which achieved an adaptive safety warning.

The design of the electric field measurement sensor is based on the capacitor principle, and its performance parameters are determined by the calibration experiments of the sensor, and it is shown in Table 4. The electric field sensor was tested at a ±500 kV DC converter station in Yunnan. The measured electric field differed from the theoretical value by 0.3 kV/m. The results show that the sensor meets the measurement requirements, which has good consistency and reliability. The innovations of the sensors developed in this article are:

(1) The measurement range and sensitivity of the sensor can be adjusted dynamically by changing the motor speed, and it is possible to measure the electric field in the space around charged devices with different voltage levels.
(2) The differential output of the sensor's sensing signal can reduce the interference of the ion field on the measured signal.
(3) The diameter of the sensor is $d = 50$ mm, which is 2–5 times smaller than the existing sensor volume on the market. The smaller sensor size can reduce the impact on the electric field distortion.
(4) The electric field sensor can be connected with the mobile phone through Bluetooth communication, so that the alarm prompt symbol and the real-time waveform of the measured electric field can also be visually seen on the mobile smart alert APP interface.

According to the spatial electric field distribution curve of the DC charged equipment, it can be known that different voltage levels correspond to different electric field gradient values. Generally, the electric field strength value near the safety distance between the device and the human body is selected as the safety threshold. If the safety threshold is exceeded, the early warning system will issue an alarm prompt. In order to verify the practicability and reliability of the early warning device, field experiments were conducted on a ±500 kV DC converter station and a ±800 kV DC transmission line experimental platform. According to the test results, it is known that the alarm distance meets the specified minimum safety distance, and there are no false positives and false negatives.

The safety early warning device in this paper still needs to be improved. The accuracy of electric field measurement needs to be further improved. In addition, the influence of electric field distortion during the measurement process is taken into consideration. The whole early warning device can only be held or left standing, which still cannot be worn. The accuracy and volume of electric field sensors need to be further optimized to develop a more reliable and practical safety warning device [32].

**Author Contributions:** Methodology, H.S.; experiment, H.S.; validation, H.S.; writing original draft preparation, H.S.; writing—review and editing, C.S.; funding acquisition, C.S.; visualization, C.S.; supervision, W.Z.; project administration, W.Z.; resources, N.Z.; data curation, W.C. All authors have read and agreed to the published version of the manuscript.

**Funding:** This research comes from the China Southern Power Grid Technology Project and the project funding was provided by the Electric Power Research Institute of Yunnan Power Grid Co., Ltd.

**Conflicts of Interest:** The authors declare no conflicts of interest.

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
