# Peer review of "Adaptive Safety Early Warning Device for Non-contact Measurement of HVDC Electric Field"

_electronics, doi:10.3390/electronics9020329_

Round 1

Reviewer 1 Report

In this paper, a high-voltage DC electric field adaptive safety early warning device based on a field-grinding sensor is developed for DC live equipment of different voltage levels. The electric field sensor used in this device which measures the electric field strength of the space around the charged equipment through non-contact measurement. The paper is well written. However, I have the following comments.

Introduction has been vaguely written. My suggestion is to divide the introduction into three subsections: 1) motivation and incitement, 2) literature review, 3) contribution and paper organization.

Language and the structure of the paper should be improved.

Editing of English language and style are required.

Explain equations 1 and 2 clearly.

Section 4.2: Adaptive Security Early Warning Algorithm needs more explanation.

Author Response

Dear reviewer,

     Thank you for your comments and suggestions. Please see the attachment.

Reviewer 2 Report

In this paper, an electric field sensor is developed which measures the electric field strength of the space around the charged equipment through non-contact measurement. However, the novelty of the paper is not properly justified as there are lots of commercial equipment available in the market. 

Moreover, the manuscript is rather suitable with this journal, preferably follows the scope to any sensor journal. Accuracy needs to be justified with other instruments. Previous work didn't highlight too much. 

Typos, errors are available e.g Table 4 in separate pages. References need to be improved.

Author Response

(The authors gave the same response as above.)

Reviewer 3 Report

1. Equation 1 is incorrectly edited.

2. In line 100 instead of "cm2" there should be "m2".

3. Unify the multiplication notation (see Eq. 6, 7).

4. The symbols in the text must be the same as in the formulas (for example see lines 98-100).

5. Delete the descriptions in Chinese shown in figures 5, 6.

6. Please cite 2-3 articles from the mdpi magazine.

Author Response

(The authors gave the same response as above.)

Reviewer 4 Report

The paper presents an adaptive safety early warning device with long-distance electric field measurement and an alarm function for DC live equipment of different voltage levels.

Manuscript’s strengths:

- The paper proposes a new type of field milled sensor which has lower power consumption.

- Experimental validation through tests performed on a high-voltage simulation platform and also on an on-site operating condition.

Manuscript’s weaknesses:

- The lack of a more detailed presentation of the proposed device performances compared to the performances of many other devices existing on the market (comparative numerical values).

Recommendations for the improvement of the manuscript:

- The abstract should be rewritten to highlights the contribution and the novelty of the paper. I don’t think that the abstract should contains numerical details.

- It should be done a more extensive and detailed comparison with other existing devices (sensitivity, error, precision, ...).  Based on the short comparison already made, only the warning function is the new feature of the proposed device? Besides the low cost and low power consumption, what are the technical advantages of the proposed solution?

- What means "early warning" in time units? Or "early" refers the safety distance?

- Can you explain what means "adaptive" safety device? It is not clear what means "adaptive" feature.

- The wireless transmission is not affected by the high voltage electrical field?

- The manuscript is quite long. I think that the Fig.7 is not relevant in the paper. Also, the algorithm from Fig.18 is too simple to be presented.

- How would Kalman filtering be used to eliminate errors caused by anomalies?

- The conclusion must highlight the novelty and the advantaged of the proposed solution.

- Regarding the English language a few minor spell checks are required.

Author Response

(The authors gave the same response as above.)

Round 2

Reviewer 4 Report

The authors responded to all the recommendations related to the review.
The manuscript has been significantly improved.